

# ADCP observations of migration patterns of zooplankton in the Cretan Sea

Emmanuel Potiris[1,2], Constantin Frangoulis[1], Alkiviadis Kalampokis[1], Manolis Ntoumas[1], Manos Pettas[1], George Petihakis[1] and Vassilis Zervakis[2]

[1]Institute of Oceanography, Hellenic Centre for Marine Research, Heraklion, Crete, 72100, Greece

[2]Department of Marine Sciences, School of the Environment, University of the Aegean, Mytilene, 81132, Greece

*Correspondence to*: Emmanuel Potiris (mpotiris@hcmr.gr)

**Abstract**. The lack of knowledge of the mesopelagic layer inhabitants, especially of those performing strong vertical migration, is an acknowledged challenge as its incomplete representation leads to the exclusion of an active carbon and nutrient pathway from the surface to the deeper layers and reversely. The vertical migration of mesopelagic inhabitants (macro-planktonic and micro-nektonic) was observed by acoustical means in the epi- and mesopelagic layer of the open oligotrophic Cretan Sea (Eastern Mediterranean) for almost 2.5 years at the site of an operational fixed-point observatory located at 1500m depth. The observed organisms were categorized in four groups according to their migration patterns. The variability of the migration patterns was inspected in relation to the physical and biological environmental conditions of the study area. The stratification of the water column does not act as a barrier for the vertical motion of the strongest migrants, moving up to 400 m every day. Instead, changes of light intensity (lunar cycle, daylight duration, cloudiness) and the presence of prey and predators seem to explain the observed daily, monthly and seasonal variability. The continuous presence of these organisms, yet capable of vertical motion and despite the profound seasonal circulation variability at the site of the observatory, implies their presence in the broader study area. The fundamental implications of the above for biogeochemical processing in oligotrophic seas due to the intimate link of the C and nutrient cycles, are discussed.

## 1 Introduction

The inhabitants of the deep sea play an important role in determining the depths to which carbon is exported, a role mainly played by microbes and zooplankton (review by Turner, 2015). However, the lack of data from midwater depths severely limits our ability to quantify the *efficiency* of the biological pump (Robinson et al., 2010). Moreover, the primary production at the surface ocean, which is the process that controls the *strength* of the biological pump, depends on nutrient cycles, and this dependence is stronger in oligotrophic seas. However, our knowledge of this process is limited in the open ocean, due to limited observations.

In the open Mediterranean Sea, and especially the oligotrophic Eastern Part, our understanding of the $CO_2$ functioning is essentially based on epipelagic observations or biogeochemical (BGC) model results. In fact, although there have been several studies on vertically migrating zooplankton in the Mediterranean Sea, little is known for its eastern part (review by Saiz et al., 2014). The scarce, short-term, observations have reported different migration strategies of zooplankton (Fragopoulou and Lykakis, 1990; Koulouri et al., 2009) and dependence upon hydrology and food availability (Nowaczyk et al., 2011).

The open Cretan Sea's biochemistry is representative of a wide area of the Eastern Mediterranean (0.6-1.6 x $10^6$ km$^2$ depending on the parameter) (Henson et al., 2016). In the Cretan Sea, the shifted seasonal cycle of primary productivity (Psarra et al., 2000) and the presence of a deep layer of high chlorophyll concentration, a common feature of the Mediterranean Sea, referred to as the Deep Chlorophyll Maximum (DCM hereafter) in the bibliography (Kimor et al., 1987; Yacobi et al., 1995), are examples of the close coupling of biological and physical processes in the broader study area. In fact, although the DCM's magnitude is mainly determined by biological mechanisms, the DCM depth and structure are essentially determined by optical-



physical factors (e.g. Mann and Lazier, 1991; Varela et al., 1994; Crise et al., 1999). The vertical extent and the intensity of the two mesoscale gyres forming a dipole in the Cretan Sea (Theocharis et al., 1999; Cardin et al., 2003; Kassis et al., 2015) govern to a large extent this coupling (Petihakis et al., 2002).

The range of zooplankton size from less than 2 µm (for heterotrophic nanoflagellates) to several meters (large jellyfish), makes

the complete representation of all its components in a single net sample difficult to achieve (Frangoulis et al., 2016 and references therein). In the Mediterranean Sea, zooplankton is generally captured with a standard 200 µm mesh size net. However, the populations of large individuals belonging to important and common predatory groups, capable of strong migrations, such as the chaetognaths, amphipods, euphausiids, decapods, have been underestimated as many of them escape the standard 200 µm mesh size net when towed at the recommend speed of <1 m s$^{-1}$ (Moriarty and O'Brien, 2013).

However, such large individuals can be detected by low frequency Acoustic Doppler Current Profilers (ADCPs hereafter). In fact, progress in ocean acoustics and marine technology during early 1980s (Holliday, 1977; Holliday and Pieper, 1980; Holliday et al., 1989; Costello et al., 1989) allowed the estimation of distribution patterns and biomass of zooplankton and micronekton to be inferred from ADCPs (Flagg and Smith, 1989). To measure currents, ADCPs transmit sound pulses in different directions. The sound is scattered by particulate matter in the water column, and radial velocities are computed from

the Doppler shift of the backscattered signal (Gordon, 1996). The intensity of the backscattered sound can also be used in conjunction with net samples for estimating the biomass of zooplankton (Ashjian et al., 2002). Although the estimated ADCP backscatter is a by-product (Bozzano et al., 2014) and thus more suitable for qualitative than quantitative analysis (Brierley et al., 1998), field studies complemented with ADCP-derived sound scattering have been used to describe biological patterns in the interior of the ocean, such as zooplankton aggregations (Zhou and Dorland, 2004) and vertical migration (Postel et al.,

2007) with remarkable detail.

During a study of the current velocities in the Cretan Sea back in 2000, using the same ADCP (75 kHz), as in the present study, deployed in the same location, Cardin et al. (2003) reported a noise in the measurements of vertical velocity. In fact, at 75 kHz objects with size of 5 mm (1/4 of transmit pulse wavelength) or more, reflect sound and thus cause a strong backscatter signal (Thomson and Emery, 2001). The migrating zooplankton was given as a possible explanation by Cardin et al. at the time.

During the present study, four consecutive deployments of a 75 kHz ADCP at the site of an operational open-sea observatory in the Cretan Sea, covering a period of two and a half years, provided a unique opportunity to study the migration patterns of zooplankton, continuously and in high frequency for a long period in relation to environmental conditions. The aim of this paper is to present the observed distribution patterns of zooplankton (focusing on diurnal migration), discuss their relation to physical and biological environmental conditions, such as daylight, currents, stratification and food resources, and to provide

new information useful for zooplankton studies in the Cretan Sea, while opening a new way to evaluate an active pathway of carbon cycle of the wider area.

## 2 Materials and Methods

### 2.1 Experimental setup

The Poseidon E1-M3A observatory (www.poseidon.hcmr.gr) is located at 25.12° E, 35.74° N in the center of the Cretan Sea

(south Aegean Sea) at a depth of 1500 m (Figure 1, a). It consists of two moorings: the first one has a surface buoy and a real-time multi-sensor array down to 1000 m, while the second is a subsurface one, hosting an upward looking RDI 75 kHz ADCP. The observing effort is complemented by monthly R/V cruises to perform CTD casts and water-plankton sampling. The ADCP data set used in the present study consist of four successive deployments of variable duration, which extended over a period of two years and seven months, from 15 November 2012 to 20 May 2015 (Table 1). The distance between the ADCP mooring

line and E1-M3A buoy varied slightly but it was less than 2.7 NM for all deployments (Figure 1, c). The primary purpose for



the ADCP deployment was to study currents. The first deployment was considered as a test of the setup, so the sampling scheme and the depth of the ADCP were selected empirically (Table 1). The first deployment confirmed the feasibility of monitoring biological activity and it became obvious that a greater depth should be chosen, since the parking depth of zooplankton was deeper than the initial ADCP deployment depth and the sea surface reflection contaminated the first 50 m of

the water column. However, the rearrangement of the mooring line was not possible due to the tight schedule of the next two deployments, and thus, only the last ADCP deployment was ~120 m deeper than the previous ones.

The ADCP sampling plan was optimized in terms of temporal and spatial resolution by setting different sampling schemes at each deployment (Table 1; Figure 1, b). The aim was to check the consistency of the vertical velocity measurements of zooplankton and backscatter coefficient (defined in Sect. 2.2) between deployments. No significant differences in the vertical

velocities and the backscatter coefficient between deployments of variable cell length and sampling rate, is an indication of reliable – accurate measurements. Thus, it is possible to identify biases caused by the sampling scheme, instead of the velocity errors due to the ADCP accuracy and of the backscatter coefficient estimation methodology.

One parameter used to characterize zooplankton behavior as well as to potentially identify optimal sampling strategies was the hereafter defined *burst velocity*. The burst velocity is defined as the maximum of the several 30-min or 1-hour velocity average

values during each ascent or each descent of the zooplankton population.

The velocity estimation inside a cell over the sampling interval is the result of the averaging of several pings. Regarding the time resolution, the challenge was to identify the lowest sampling rate that would still give acceptable resolution of the ascending/descending zooplankton movement, while conserving power and extending the deployment period as much as possible. Two sampling and averaging intervals, of lengths 30 min and 1 hour respectively, were tested in order to select the

optimum sampling scheme. During the first deployment, a sampling interval of 30 min was used, to be followed by a 1 h interval during the second deployment. Comparison of the data from the two deployments revealed an underestimation of burst migrating velocities during the second data set because of the lower sampling rate (1 h), thus the initial value of 30 min was selected for the last two deployments.

The range of the cells used (10 – 20 m) did not affect the burst velocity and the average velocity measurements. However,

using a small bin size (10 m) during the last deployment resulted in noisy velocity measurements. The depth-integrated $S_v$ (backscatter coefficient, defined in Sect. 2.2) between the depths observed at all deployments were also consistent. The seasonal variability of the physical properties of the water column affected the estimation of $S_v$ at depths shallower than 100 m, mostly during late August. Placing the ADCP at an upward looking position at a smaller depth than the nominal range resulted in erroneous data of the first 50 m of the water column due to sound reflection on the sea surface.

**2.2 Data processing/analysis and visualization of Backscatter data**

The backscatter coefficient $S_v[\text{dB re }(4\pi\text{ m}^{-1})]$ is given as:

$$S_v = C + 10\log_{10}\big((T_x + 273.16)R^2\big) - L_{DBM} - P_{DBW} + 2aR + K_c(E - R_r), \tag{1}$$

where $C\,[\text{dB}] = -159.1$ is an instrument constant, $T_x[°\text{C}]$ the transducer temperature, $R\,[\text{m}]$ the slant range, $L_{DBM}$ the $10log_{10}$ of the transmit pulse length $[\text{m}]$, $P_{DBW}$ the $10log_{10}$ of the transmit power $[W]$, $a[\text{dB m}^{-1}]$ the sound absorption coefficient,

$K_c$ a constant of proportionality for converting the incoming raw echo data to dB, $E\,[counts]$ the raw echo data and $E_r = \min(E)\,[counts]$ the reference raw echo per transducer when there is no signal. $S_v$ was calculated according to Deines (1999), $K_c$ according to Heywood (1996), the speed of sound for the calculation of $R$ according to Gordon (1996) and $a$ according to Ainslie and McColm (1998).

Instantaneous vertical velocity profiles were depth averaged and split in daily pieces to identify the hours of the day during

which the zooplankton moves upward or downward, as well as the long-term variability in the ascend/descend hours. Another step was to select the maximum and minimum vertical velocities of each daily piece of data during the time of the upward and





downward movement. Depending on the sampling rate, two to four samples were averaged. At last, histograms of vertical velocity versus depth one hour before and one hour after sunrise/sunset times were used to identify possible ascending/descending differences of migration patterns and evaluate the consistency between the different ADCP sampling schemes.

Climate Data Operators (CDO, 2018), Ocean Data View (Schlitzer, 2016) and Generic Mapping Tools (Wessel et al., 2013) were used for the data processing and visualization. Wind stress and sensible/latent fluxes were computed from the quality-controlled buoy data with the air-sea toolbox to identify the time when conditions favor overturning of the water column (http://woodshole.er.usgs.gov/operations/sea-mat/air_sea-html/index.html).

### 2.3 Description and processing of auxiliary data

To estimate volume backscattering, assess environmental conditions during deployments and assist interpretation of ADCP measurements, several complementary data sets have been used (Table 2).

The E1-M3A buoy measures meteorological variables (wind speed, gust and air temperature were used here) as well as temperature, conductivity and fluorescence at 20, 50, 75 and 100 m. Temperature and conductivity measurements are also available at the sea surface and at 250 m. A downward looking Nortek 400 kHz ADCP is mounted to the buoy hull, measuring

horizontal currents at 5 m bins from the surface down to 50 m. Due to the lack of compatibility of the 400 kHz ADCP with the buoy software, backscatter measurements are not available from this instrument. The above meteorological and marine surface parameters were downloaded from the Poseidon on-line database (www.poseidon.hcmr.gr), where they are stored in real-time. In addition, due to occasional problems with the real-time underwater transmission, subsurface sensor data were downloaded from the memory logs of the instruments during the regular bi-annual maintenance. Meteorological and sea surface

measurements from the buoy's sensors span a period of 24 M (from 22 May 2013 to 25 May 2015) and subsurface measurements a period of 20 M (from 22 May 2013 to 10 January 2015). Buoy data have undergone automated quality control, such as rejection of stalled values and application of min-max and spike filters. Visual inspection was the last quality control step; the remaining suspect measurements were removed manually. The heat flux through the air-sea interface computations were based on the air-sea interaction Matlab routines provided by Rich Pawlowitz (via the SEAMAT collection, https://sea-

mat.github.io/sea-mat/), applied on the E1-M3A meteorological and sea-surface data.

At the E1-M3A zooplankton samples are taken regularly (monthly) with parallel tows of two nets (45 μm mesh size and 200 μm mesh size) from 100 m to the surface. To validate the ADCP measurements a field sampling strategy targeting zooplankton organisms at the size of 5 mm would require day and night multilayer net tows down to 500 m depth, with mesh size >200 μm towed at a speed >2 NM. Unfortunately, logistically this proved not feasible and only on 19/12/2013 at 12H00 (local time)

one oblique tow at a speed of 2 knots with a Bongo net equipped with a 330 μm and a 500 μm net (both with a flowmeter) was performed. As this was possible only once, the information obtained is considered as an indication of the type of large metazoans present above the ADCP at the sampling time, but cannot be considered representative of the average relative abundance of migrants in the area and thus mentioned briefly in the discussion.

## 3 Results

### 3.1 Environmental conditions at the study site

The ADCP site is located at the center of the semi-permanent dipole of the Cretan sea, which consists of a cyclone to the east and an anticyclone to the west of the observatory (Theocharis et al., 1999; Korres et al., 2014). Four water masses fill the surface and subsurface layers of the Cretan sea. Modified Atlantic Water (MAW, S=38.5-38.9 psu) fills the 20 m-100 m layer. Cretan Intermediate Water and Levantine Intermediate Water, that have similar characteristics (CIW & LIW, θ=14.9-15.1 °C,



S~39-39.1 psu) fill the 200 m-500 m layer. Transitional Mediterranean Water (TMW, θ=14.2 °C, S=38.92 psu), a mixture of Levantine Intermediate Water and Eastern Mediterranean Deep Water enters through the Cretan Straits and its core lies at the 500 m-800 m layer (Georgopoulos et al., 2000; Velaoras et al., 2013) or deeper (Velaoras et al., 2015). Below TMW lies the Cretan Deep Water, a water-mass argued to have local (Theocharis et al., 1999) or North/Central Aegean origin (Zervakis et

al., 2000; Gertman et al., 2006). The temperature at the depth of the deep scattering layer ~450 m (based on the available data set; Figure 2, d, e & f), ranges from 14.55 °C to 14.9 °C and the salinity from 38.98 psu to 39.04 psu.

Sea surface temperature ranges seasonally from 15 ºC to 26 ºC and salinity ranges from 38.8 psu to 39.5 psu (Figure 2, b, c). The salinity of the deeper layers ranges from 38.9 psu to 39.1 psu. Lowest temperatures are observed during February and March, while highest temperatures during August and September. The seasonal cycle of temperature penetrates down to 100

m and the permanent thermocline extends down to 350 m (Figure 2, b). Salinity also exhibits a seasonal cycle down to 100 m but the seasonal signal dominates salinity variations of the upper part of the water column (Figure 2, c). Largest salinity values are observed during calm, cloud free summer days, due to intense evaporation and the intrusion of high salinity surface water of Levantine origin (Theocharis et al., 1999). Inflow of Atlantic water (Theocharis et al., 1999), typically during late summer, causes the salinity minimum at the subsurface layer from 40 m to 100 m in Figure 2, c & e. Deep casts (Figure 2, e & f) reveal

a continuous change of the water column towards fresher and colder values between 250 and 1000 m especially from 2012 to 2016, which points to intensified horizontal motion of the subsurface layers.

At the study area, prevailing winds blow from N-NW (Figure 3, a). Short-term variability of air temperature during winter is larger, due to strong northerly winds, which cause the air temperature to drop below 10 ºC. Latent heat loss typically ranges between 100 Wm$^{-2}$ and 200 Wm$^{-2}$. Sensible heat flux also results in net loss, but it is negligible from March to October. During

the rest of the year typical values are less than 50 Wm$^{-2}$. Sensible heat loss contributes significantly to the flux budget when wind stress becomes larger than 0.2 Pa (typical values are 0.1 Pa). Wind stress during December of 2013 was more than 0.2 Pa on average, while peak values of 0.8 Pa were also observed. Consequently, the monthly average sensible heat flux was about 100 Wm$^{-2}$ and peak values were about 300 Wm$^{-2}$. Latent heat during that period peaked at 600 Wm$^{-2}$. Similar atmospheric conditions that favor convection of the upper 100 m of the water column were observed during 10 to 22 February 2015, a

period during which changes in the vertical distribution of zooplankton were observed.

Average water velocity from surface down to 50 m is 0.29 ms$^{-1}$ towards S-SE and is invariant with depth. The layer between 50 m and 350 m is characterized by a diminishing vertical shear that is largest between 50 m to 150 m and vanishes below 400 m (Figure 3, d, where only the fourth deployment is displayed, since in previous deployments the larger bin size caused underestimation of the high vertical wavenumber shear). The average current speed below 350 m is 0.06 ms$^{-1}$. The direction

of the axis of maximum variance between the surface and 50 m is S-SE and gradually turns to S-SW at 200 m. Currents are less unidirectional with depth too. The strong currents of the surface layer exhibit the least directional variability (Figure 3Figure , b & c). High frequency variability at the site consists of inertial and tidal currents, which account for a small portion of the total variance (less than 8 %), even though the inertial motions are dominant over short periods. Low frequency variability is controlled by the intensity and the vertical extent of the dipole as Cardin et al. (2003) have mentioned.

During the study period the core of DCM was observed between 70 m and 120 m and its vertical extent was ~60 m (Figure 4, a). On average, the largest chlorophyll values are observed at the 75 m and 100 m sensors of the buoy (Figure 4, b). Also, at these depths, the short-term variability is comparable to the variability due to the annual cycle, while for the 20 m and 50 m sensors the seasonal variability is dominant. The DCM starts to form from February to April and is usually destroyed by October (see changes of the depth range for which chlorophyll concentration is above 70 % of the maximum value in Figure

4, a, as well as Figure 4, b).



### 3.2 Scattering and migrating groups of organisms

The results of the four deployments of a total duration of 2.5 years (Figure 5; Figure 6) provide a wealth of information about the scattering organisms and their movement in the water column. Characteristics that are easily visible include the constant presence of a deep layer of scatterers (unfortunately visible only in the last deployment, since only then was the ADCP placed

deep enough to record it), a diel, a seasonal and a monthly (moon) cycle.

A closer examination of daily backscatter patterns (Figure 6), allows the categorization of the scattering organisms in four groups according to their migration patterns, on the basis of distinguishable trails of volume backscatter measurements of the ADCP. Three of them exhibit a daily migrational pattern, while the fourth remains at a constant depth. The first group (group A hereafter) does not migrate. It is found at 400 m - 450 m, and it forms a permanent deep scattering layer (Figure 5 last

deployment, Figure 6). The width of this scattering layer varies, signifying that it does not consist only of scatterers of group A, but there are also migrating organisms, which have their parking depth there, and spend part of their time in other depths.

Group B (Figure 6) follows the normal daily vertical migration pattern i.e. it moves close to the surface at dusk, feeds there during night and returns to the parking depth at dawn where it stays during the day. This group spends the day-time at 400 - 450 m and the night-time between 150 m and surface (Figure 5, last deployment). When at the bottom of the seasonally varying

feeding layer (60 - 160 m) (Figure 6), its vertical velocity decreases, probably as a result of feeding activity, while still moving towards the surface. The bottom of the feeding layer is identified by the deceleration of upward movement and subsequent increase of $S_v$ (Figure 6), as the zooplankton spends more time in a particular cell when moving at smaller speed. The change of depth of the bottom of the feeding layer of group B is in good agreement with the time variation of the depth of maximum chlorophyll concentration (Figure 4).

The backscatter coefficient at any certain depth, as long group B is above that depth, is larger during night-time compared to day-time (Figure 6). The exception to this rule is the deep scattering layer. The result is the "curtain" shape seen in Figure 6, which implies that a part of zooplankton that forms group B spreads in the entire 0 - 400 m water column while migrating. The smallest $S_v$ values, close to the system noise floor, are observed between 250 - 300 m, when group B is found at the parking depth.

Between the depth of 200 m and 250 m, $S_v$ never falls close to the noise floor, even in the absence of group B during day-time, pointing to the presence of scatterers and thus to a third group of organisms (group C hereafter). Group C also migrates from 350 m to 300 m or from 250 m to 200 m depending on the period of the year, probably as result of the change in day-time duration.

In shallower depths a fourth group can be observed (Figure 6, Group D), which spends most of the day-time in a depth of 180

m - 240 m and during the night it moves to more shallow depths of 60 m - 90 m, where its trails meets with those of group B. This is close to the depth where the layer with the largest concentration of phytoplankton throughout most of the year (Figure 4, 5) is observed. Its backscatter signal is not as strong as that of group B. Actually, its trail is easily distinguishable mostly during the time of its upward motion, as a secondary thin strong $S_v$ trail, shallower than the one caused by group B, and less during the downward motion (Figure 6). This signal is present at all deployments, and its characteristics, such as depth and

slope, are consistent between deployments. With close examination, we can exclude the possibility of a false/mirroring echo coming from group B, as the changes of the trails of group B are not reflected in the trails of group D. Moreover, the profiles of vertical velocities are distorted towards lower speeds at the depth where group D is observed, from 180 m to 240 m (Figure 7, a & b; Figure 9, all panels), while profiles of the time average $S_v$ between 180 m and 240 m is also larger.

An interesting outcome is that the depth of the feeding layer is also significantly affected by the phase of the moon. Full moon

is accompanied by the deepening of the bottom of the feeding layer by more than 50 m as can be seen in the backscatter coefficient (Figure 5, mainly in months October to February) and vertical velocity graphs (Figure 7, a & b).

An interesting observation derives from the combination of variability of the bottom of the feeding layer which is mainly on monthly basis, with the variability of the parking depth which is only on seasonal basis. This was evident at least for group B





which migrates a larger distance. In fact, the parking depth of group B changes seasonally following the variation of the feeding layer, however, it does not present a monthly variation (Figure 5). This implies that the distance that group B travels changes more on a monthly basis than on a seasonal basis. This is consistent with the monthly variation of the migrating velocities seen in Figure 7.

The distinction of group B from group A appears from the daily variation of the thickness of the deep scattering layer (Figure 10, e). It becomes thinner and the backscatter coefficient at that depth becomes larger during day-time compared to night-time. We infer that group A (although remaining on average at the same depth) spreads during the night, taking advantage of the absence of group B and aggregates again during day. Thus, during the short time period that group B leaves the parking depth the whole deep scattering layer spreads. Then, after the two groups are separated and for as long as group A is separate from

group B, the deep scattering layer progressively becomes thicker. As soon as group B returns to the parking depth, the deep scattering layer, now consisting of both groups, shrinks, becomes denser and thus its backscatter coefficient increases. This would be an expected behavior if group B preys on the migrating zooplankton that forms group A. Although it is not possible to synthesize a clear picture from the available data, the persistent observations that the deep scattering layer becomes thicker by 30 m to 50 m during night supports the above hypothesis. Although the deep scattering layer was only partly in the range

of the ADCP during most of the period of the 4th deployment, during March and April 2015 the scattering layer was shallow enough to be properly measured by the ADCP (Figure 5; Figure 10, e). A contiguous vertical motion of the deep scattering layer in addition to the diurnal spreading was observed. The extent of the vertical motion was ~100 m during March and ~50 m during April 2015 (Figure 10, e).

**3.3 Migration timing, duration and velocity**

Concerning the three migrating groups (B, C, D) there are no differences in the duration between fast upward and downward movement, with two hours spent each way (four hours in total) (Figure 8, a). The duration of migration is roughly the same for all migrating groups (B, C and D), as they start to move upwards at the same time (with B, D trails meeting at the feeding layer), although it must be mentioned that only four measurements are available during each upward/downward motion. The upward motion of group B begins at its parking depth of ~350 m and ends at the bottom of the feeding layer (Figure 7, a;

Figure 9, b, d, f & h). The depth of maximum negative acceleration, which marks the beginning of downward motion (the velocities of groups B, C and D cannot be distinguished), is generally found shallower than the bottom of the feeding layer (Figure 9, a, c, e & g), suggesting that zooplankton moves shallower as it feeds. Descend is symmetrical with respect to the sunrise. It starts one hour before and ends one hour after the sunrise (Figure 8, a). Ascend starts half an hour before and ends one and a half hour after the sunset (Figure 8, b). As mentioned, it is not possible to distinguish the velocity of the migrating

groups B and D, since the velocity measured by the ADCP above 250 m is the result of the motion of both groups. Depth-averaged velocities during migration time are ~3 cms$^{-1}$ (Figure 9, a).
Group B does not migrate at a constant velocity (Figure 7; Figure 9). Largest upward velocities, ~6 cms$^{-1}$ are recorded between 200 m and 300 m. Largest downward velocities are recorded between 250 m and 350 m (Figure 9). Since Group B travels the largest distance in the course of a day and goes deeper than group D, the largest vertical velocities recorded, especially between

200m and 350m, must be due to the migration of group B. At the depth of ~200 m the ADCP records relatively small vertical velocities, about 2 cms$^{-1}$ (Figure 7, a & b; Figure 9, all panels), which distort the vertical profile that would be expected by group B, unless group B decelerates at the bottom of the photic zone. The dispersion of the vertical velocity around the average value at that depth is much less than all the other depths. While the cause is clearly biological, to date it was not possible to understand and explain it using the available data, though the spreading of group B and the presence of group C that prays

around that depth could explain these observations to some extent.



At 35° longitude, day-time lasts 9.8 h on winter solstice and 14.5 h on summer solstice. So, the time interval that the zooplankton has at its disposal in order to feed varies significantly in the course of a year (Figure 8, a). This affects the velocity of group B, which changes seasonally as is clearly shown in the burst velocities of Figure 8, b, despite the fact that they are quite noisy. The phase of the seasonal cycles of burst speeds and day-/night-time almost coincide. Downward velocity is

slightly larger than the upward velocity by ~1 cms$^{-1}$ on average (Figure 8, b). The largest burst speeds are recorded during spring, when the seasonal pycnocline starts to form. This is the period of the year that the phytoplankton (chlorophyll-α) is spread quite homogeneously throughout the upper 160 m of the water column and the DCM is not formed yet (Figure 6). Thus, the seasonal changes of velocity are driven by the combination of short night-time and low food resources found at a relatively small depth, the duration of night-time being the governing factor.

**3.4 Effect of an extreme meteorological event**

Three successive events of harsh weather conditions were observed from 10 to 13, from 17 to 21 and from 23 to 25 February 2015. The sky was mostly overcast (Figure 10, a), air temperature dropped at 7.5 °C (Figure 10, b), wind speed reached 15 ms$^{-1}$ and wind gust exceeded 20 ms$^{-1}$ (Figure 10, c). The third event was shorter than the first two and caused an increase in air temperature. The homogenization of the water column prior to the first event did not exceed the 50 m (as shown from the E1-

M3A time series) while the nearest in time available CTD cast on the 3$^{rd}$ of March revealed that the first 100 m of the water column were then homogenized. The zooplankton was distributed from the surface down to 350 m all day long (Figure 10, e), especially during the first two events, although $S_v$ remained larger during night-time compared to day-time above the depth of 300 m. Also, only small migrating velocities were measured during the events (Figure 10, d). During the second event, the core of the deep scattering layer moved shallower at the depth of 350 m. After the third event, the pattern of the backscatter

coefficient above 300 m returned back to the "normal" conditions, but the deep scattering layer remained generally shallower and moved coherently in the vertical direction from 450 m to 350 m until the 2$^{nd}$ of April (Figure 10, e). Thus, the combination of low light level due to clouds and convective currents, triggered changes at the deep scattering layer, which is found well below the maximum depth at which the overturning took place. Cloudiness may have an indirect effect on migrants, since the phytoplankton production, and thus the available prey concentration, becomes lower under lower light conditions. In addition,

the prey is spread downward due to convection. Thus, the migrants have to spread in a larger water column in order to obtain a sufficient amount of prey.

**4 Discussion**

Migration patterns of zooplankton have been observed by an acoustical method using a long-term time series for the first time in the eastern Mediterranean. The presence of organisms >5 mm, that backscatter the signal of a 75 kHz ADCP was recorded

from the surface down to ≈ 450 m. The persistent observations of these organisms throughout the duration of all deployments at the same location, even though the circulation in the study area is quite variable (e.g. moving semi-permanent cyclone-anticyclone dipole, Korres et al., 2014), suggests the widespread presence of different groups of organisms >5 mm in the Cretan Sea. These organisms were categorized in four groups according to their migration behavior, with three of the groups exhibiting vertical migration. A non-migrating group (group A) that shares the 450-m horizon with two migrating groups (B

and C) was found. A third migrating group (group D) parked at an average depth of 180 m. Similar results were obtained in the Arabian Sea using a 153 kHz ADCP by Luo et al. (2000) who observed two groups migrating simultaneously. The vertical distance between the migrating groups observed by Luo et al. (2000) was larger during the upward motion and smaller during the downward motion, which might explain why the 75 kHz ADCP used in this study could not distinguish the two of the migrating groups (groups B and D) during the downward motion. Average migrating velocities of ≈ 3 cms$^{-1}$ and burst velocities

of ≈ 6 cm s$^{-1}$ were measured corresponding to the first migrating group (B) and agree with the average and burst speeds



reported in Ott (2005) and references therein. Migrating velocities of groups C and D appeared to be smaller than those of group B. Interestingly, the migrating velocities obtained were the same, whether the velocity was directly measured by the ADCP, or via the indirect calculation using the backscatter signal (350 m migration over 2 hours, i.e. $\approx 5$ cm s$^{-1}$).

Daily dispersion, aggregation and small vertical motion of the deep scattering layer were observed. The examination of zooplankton aggregation is out of the scope of this study due to the necessary compromise between sampling rate and total deployment duration, which result in coarse data for this purpose. However, the diel vertical migration and the avoidance of predation are well known biological drivers of the intensification of patchiness (Folt and Burns, 1999). The spreading out of the non-migrating group (A) at night during the absence of the migrating group (B), and its shrinking back during day-time during the presence of B, suggests that these two groups are coupled in a behavioral relation.

Diel vertical migration of zooplankton has been related to several exogenous and/or endogenous factors (review by Ringelberg, 2010). In the present study light intensity was found to be the major factor affecting the zooplankton migration. Seasonal and monthly variability, evident in the backscatter coefficient and vertical velocity, was dictated by the duration of day-time and moon phase respectively. Twilight effects on migration patterns using data from a downward looking 300 kHz ADCP measuring from the surface down to 80 m were also reported by Bozzano et al. (2014) in the Ligurian Sea, while smaller

amplitude changes of the extent of vertical migration due to changes in cloudiness can also be found in the results of Pinot and Jansá (2001). According to the results presented here, during full moon the zooplankton preys almost 50 m deeper than during the new moon, a possible behavioral response to increased light conditions.

Another factor affecting migration was prey (in terms of Chl-α) concentration and location. The seasonally varying zooplankton feeding layer extended from the surface down to a maximum depth of 160 m. The bottom of the feeding layer

was found at an average depth of 100 m. It was recorded deeper from May to July and shallower from November to January. The upward motion of the migrating groups decelerated at the depth of the largest chlorophyll concentration. The observations suggest that the vertical gradients of temperature, salinity, density and horizontal currents affect the migration of zooplankton less than their feeding behavior.

The fact that the parking depth of the migrating zooplankton groups B and C (which is also the parking depth of the non-

migrating group too) is found so deep (450 m), cannot be explained by light, phytoplankton prey concentration (since these are zero below 200 m), nor by a temperature, salinity or density gradient at that depth. Considering that at the parking depth of these groups the vertical shear practically vanishes and the horizontal currents are the weakest ones recorded, it might indicate an active behavioral adaptation to minimize energy loss by maintaining their position at a depth with minimum turbulence.

The combined effect of wind, cloudiness and convection in the upper layer affected the migrating groups by reducing their vertical velocities. They spread in almost the entire water column, and did not migrate as deep as usual during day-time. While convective events were not observed down to 450 m, the deep scattering layer was uplifted by several tenths of meters and its daily vertical motion became larger. Shoaling of the deep scattering layer was observed during winter and deepening during summer. The overall picture is that the environmental conditions affect the migrating groups and the changes propagate to the

non-migrating group.

Several limitations of the ADCP and auxiliary data should be carefully considered. $S_v$ is a proxy for zooplankton biomass and when integrated along the acoustical beams it can provide a gross measure of the instantaneous biomass of the water column (changes in the acoustical character of zooplankton cannot be identified). Although the integrated $S_v$ is consistent among the deployments (discrepancies between deployments were observed only for the first few bins), such an analysis is not meaningful

with the experimental configuration of this study, since the zooplankton is not permanently in the range of the ADCP and because of the seasonal succession of dominant species constituting the zooplankton stocks in the Cretan Sea (Gotsis-Skretas et al., 1999). The upper 50 m of the water column are not measured, thus the depth integrated $S_v$ exhibits significant variability due to the monthly change of the depth of the feeding layer because of moonlight. Also, the whole deep scattering layer is



found inside the ADCP range only for a small period of the 4[th] deployment, adding another source of variability that is not attributed to biomass changes of zooplankton. Another source of error, which largely depends on the availability of auxiliary data, is the imperfect calculation of the effects of the gradients of the upper water column in the estimation of $S_v$ due to the changes in temperature and salinity. The above problems are generally encountered when measuring zooplankton with upward

looking ADCPs and should be treated with caution in acoustical studies of zooplankton.

Another limitation is due to the insufficient in situ data from large zooplankton in the area. Local literature does not allow us to clearly identify the taxonomic composition of the migrating populations found in the present study. The few published studies that have sampled zooplankton in the epipelagic and mesopelagic layer of Cretan Sea, were all done with vertical hauls ($\approx 1 \, \text{m s}^{-1}$) of 200 μm mesh size nets (Mazzocchi et al., 1997; Siokou-Frangou et al., 1997; Siokou et al., 2013) and are ongoing

similarly during the monthly monitoring program at the E1-M3A observatory. Thus, they are inappropriate to capture organisms at the size of 5 mm, which are the smaller organisms expected to contribute significantly to the backscatter of a 75 kHz ADCP. It is however clear that the populations examined in our study include organisms other than copepods, since the biggest copepods species reported in the area (Mazzocchi et al., 1997; Siokou-Frangou et al., 1997; Siokou et al., 2013) reach a maximum size of $\approx 3.5$ mm (Razouls et al., 2005 – 2018). The only qualitative indication about the nature of these migrators

in the Cretan Sea is one tow made above the ADCP in December 2013, that captured large organisms (>5 mm) from which known migrators were decapod larvae, euphausiid larvae, siphonophores and chaetognaths (Figure 11). Indications can also be given by studies targeted on zooplankton migrators in the Western Mediterranean Sea by Andersen and collaborators (Andersen and Nival, 1991; Andersen et al., 1992 a & b, 2001; Sardou et al., 1996). Among the several migrant species reported, the most abundant ones that were present all year round (euphausiids, siphonophores and decapods) were

concentrated above 150 m at night-time, whereas during day-time the depth of their maximum abundance was found seasonally variable (between 300 m and 500 m) (Sardou et al., 1996). These groups appeared to have similar behavior to group B in the present study. Small euphausiids migrated from 420 m to 240 m, whereas non-migrants remained below 300 m (Sardou et al., 1996), with similar behavior to groups C and A respectively in the present study.

The above work reveals a significant problem associated with the in situ sampling of the above mentioned zooplanktonic

groups. Considering that the clear majority of samplings in the area take place during day time, above 100 m depth (when groups A, B, C, D are at the deeper parts) and with inappropriate net type and tow to capture large organisms (as explained above), it is rational to assume that they are misrepresented in the samples. Appropriate sampling strategy with day and night sampling regularly (monthly frequency) with appropriate net type and tows to study diel and seasonal variation of large organisms, has been done in few locations such as the Ligurian Sea (Sardou et al., 1996), the ALOHA site (Al-Mutairi and

Landry, 2001) and the BATS site (Madin et al., 2001; Jiang et al., 2007), with significant logistical effort.

If large stocks of large zooplankton actively migrate over significant vertical distances, in an oligotrophic deep system such as the Cretan Sea, then, new carbon pathways will have to be included in our models, reconsidering the energy flow and the dynamics of the system. In fact, since the carbon inflow (feeding) to the migrant groups comes from lower trophic levels (i.e. phytoplankton) at the euphotic zone, the zooplankton migrators may cause an important active downward vertical flux of

matter, thus increasing the biological pump's efficiency (review by Frangoulis et al., 2004). The lack of data from midwater depths severely limits our ability to quantify the efficiency of the biological pump (Robinson et al., 2010). In the Cretan Sea, the lack of knowledge of the role of zooplankton vertical migration and the functioning of the whole mesopelagic ecosystem may constitute an important knowledge gap of the biological pump's efficiency in the area that requires exploration. Additionally, the observed patterns are expected to have significant implications in the system dynamics particularly if one

considers the oligotrophic character of the Cretan Sea. The observed migration is expected to act as a transfer mechanism of organic matter (carbon and nutrients) from the euphotic zone to the deeper parts of the water column, overcoming the physical barrier of the pycnocline. This active flux of matter may occur since the migrations speed recorded (>3 cm s$^{-1}$) were higher than reported zooplankton faecal pellet sinking speeds (<1 cm s$^{-1}$ for euphausiids- review by Frangoulis et al., 2004). This





mechanism will enhance the oligotrophism of the mesopelagic layer since there are no effective mechanisms of very deep-water mixing – there is a strong decoupling of the surface layers with the deeper parts of the water column. Thus, the surface layers are deprived of important nutrients, although in the actual nutrient budget one has to take into account other parameters such as zooplankton excretions at the surface layers etc.

**Acknowledgements**

Part of the work has been funded by JERICO-NEXT project. This project has received funding from the European Union's Horizon 2020 research and innovation programme under grant agreement No 654410.

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





**Table 1: The deployment parameters of the upward looking 75 kHz RDI ADCP on the subsurface mooring line of E1-M3A are listed.**

| Deployment | Start | End | Bins | Bin size (m) | Sampling interval (s) | 1st Bin (m) | Average depth (m) |
|---|---|---|---|---|---|---|---|
| 1st | 15-Nov-2012 | 23-May-2013 | 25 | 16 | 1800 | 24.59 | 369 |
| 2nd | 01-Jun-2013 | 19-Jan-2014 | 33 | 12 | 3600 | 20.65 | 383 |
| 3rd | 19-Jan-2014 | 10-Oct-2014 | 25 | 20 | 1800 | 28.58 | 370 |
| 4th | 10-Oct-2014 | 02-Jun-2015 | 45 | 10 | 1800 | 18.76 | 509 |

**Table 2: The type, source, time coverage and resolution of the auxiliary data are listed. In situ data have gaps of variable length. Monitoring by R/V refers to the monthly monitoring program by regular R/V visits at the E1-M3A observatory site. NASA refers to the Goddard Space Flight Center.**

| Parameter | Type | Source | Time Coverage | Time Resolution |
|---|---|---|---|---|
| Air temp & wind | *In situ* | E1-M3A buoy | 2013/05-2014/10 | 3 h |
| Surface currents (0-50 m) | *In situ* | E1-M3A buoy (ADCP 400kHz) | 2013/05-2015/05 | 3 h |
| Subsurface currents (0-400 m) | *In situ* | ADCP (75kHz) | 2012/11-2015/05 | 0.5-1 h |
| Water temp & sal | *In situ* | E1-M3A buoy | 2013/05-2015/01 | 3 h |
| | *In situ* | Monitoring by R/V | 2010/03-2015/01 | 1 m |
| | Reanalysis | SeaDataNet | Climatology | 1 m |
| pH | Reanalysis | SeaDataNet | Climatology | 1 m |
| Chl-α | *In situ* | E1-M3A buoy | 2013/05-2014/06 | 3 h |
| | *In situ* | Monitoring by R/V | 2010/03-2015/05 | 1 m |
| Cloud fraction & optical thickness | Satellite | NASA | 2015/02-2015/03 | 1 d |





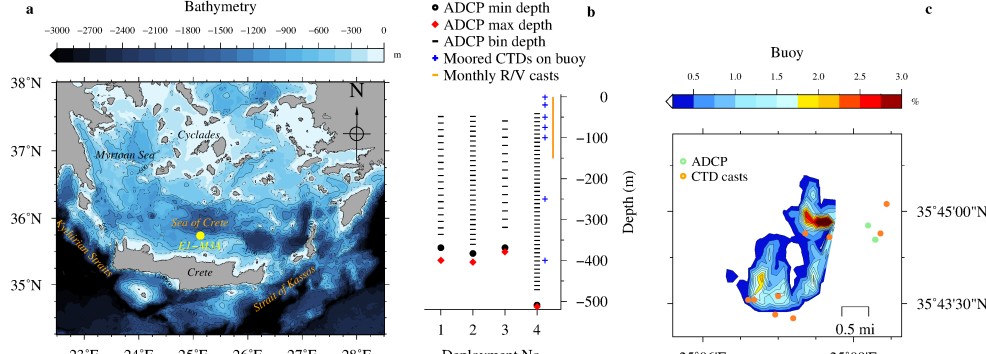

**Figure 1: Topographic map of the south Aegean Sea (a), vertical (b) and horizontal (c) views of the sampling set up at E1-M3A. Details of ADCP deployments are given in Table 1. Horizontal buoy motion is shown as the percent of the time of total deployment duration spent at a location.**



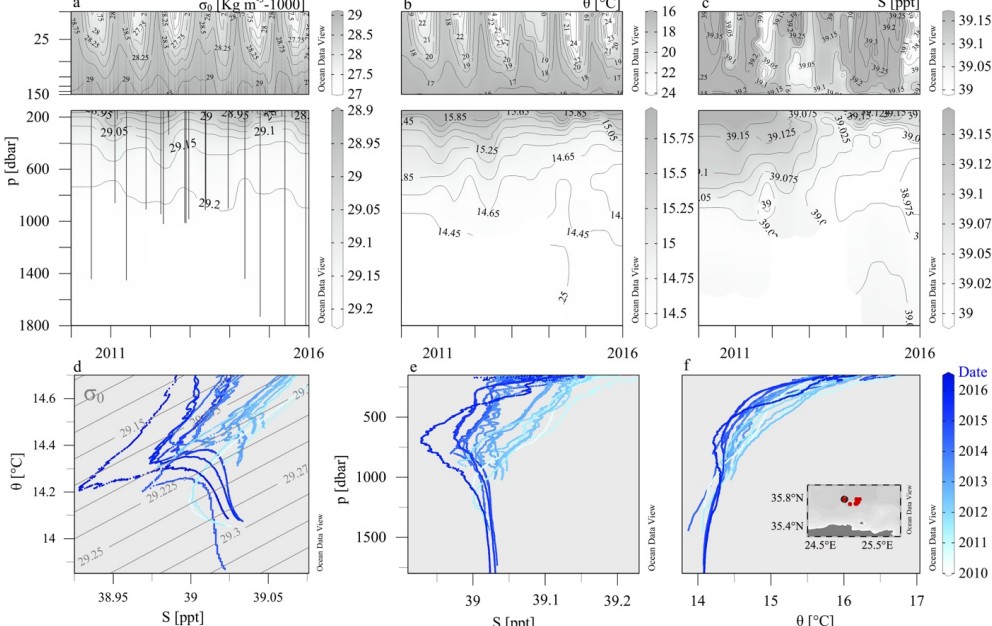

**Figure 2: CTD casts collected during monitoring and maintenance visits at the site of E1-M3A observatory with HCMR research vessels Aegaeo, Philia and Iolkos from 2010 to 2016. Time-depth plots of potential density anomaly $\sigma_0$ (a), potential temperature $\theta$ (b) and practical salinity $S$ (c). $\theta - S$ plot (d) and vertical profiles of $S$ (e) and $\theta$ (f) are colored according to date to reveal temporal trends.**



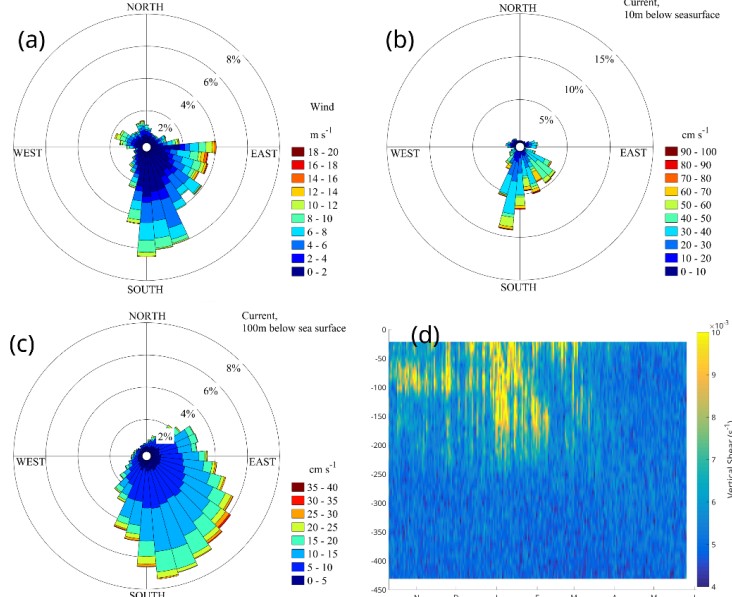

**Figure 3: Rose diagrams of wind (a) and surface (10 m depth) currents from the buoy's ADCP (400 kHz downward looking) (b), subsurface currents at 100 m depth from the upward looking 75 kHz ADCP (c) and vertical shear from the fourth deployment (d). The direction in panels (a, b, c) point to the direction of the flow.**



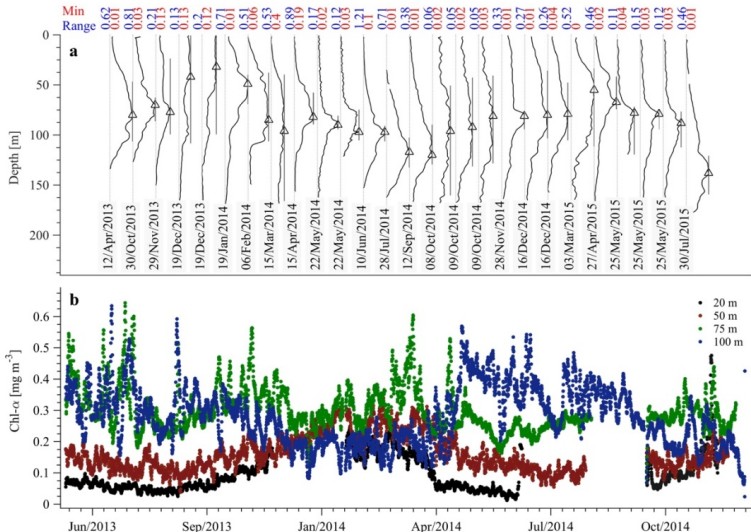

**Figure 4: Chlorophyll concentration from the CTD casts (a) and E1-M3A CTD sensors (b). The casts show the vertical distribution of chlorophyll concentration in the water column (normalized, solid black lines). The minimum value of each cast is denoted by the vertical grey dotted line and the maximum value of each cast is denoted by the black triangle. The gray bars around the triangle denote the depth range for which the chlorophyll concentration is above 70 % of the maximum value of the cast. The minimum value and the range of original chlorophyll values (in mg m$^{-3}$) are shown above each cast in red and blue colors respectively. E1-M3A chlorophyll data are low passed with a one-day running mean filter.**





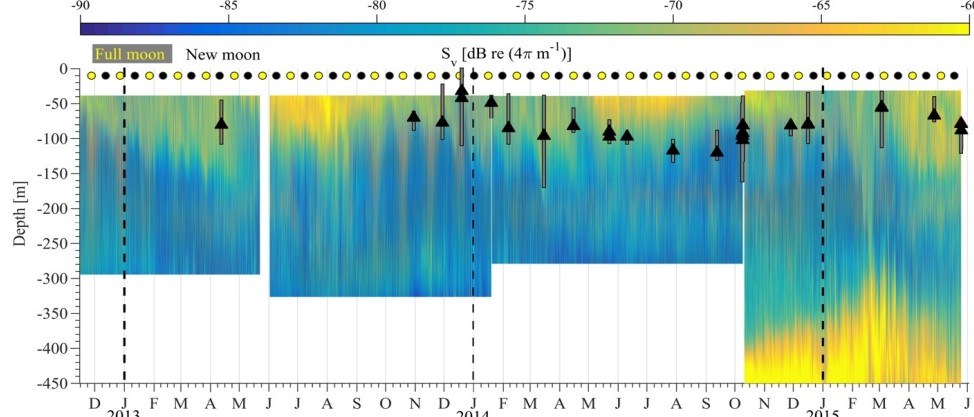

**Figure 5: The backscatter coefficient for all ADCP deployments is shown. The beginning of a year is denoted by the dashed vertical line. The black and yellow circles denote the dates of full moon and new moon respectively. The black triangles and the gray bars have the same meaning as in Figure 4, a.**





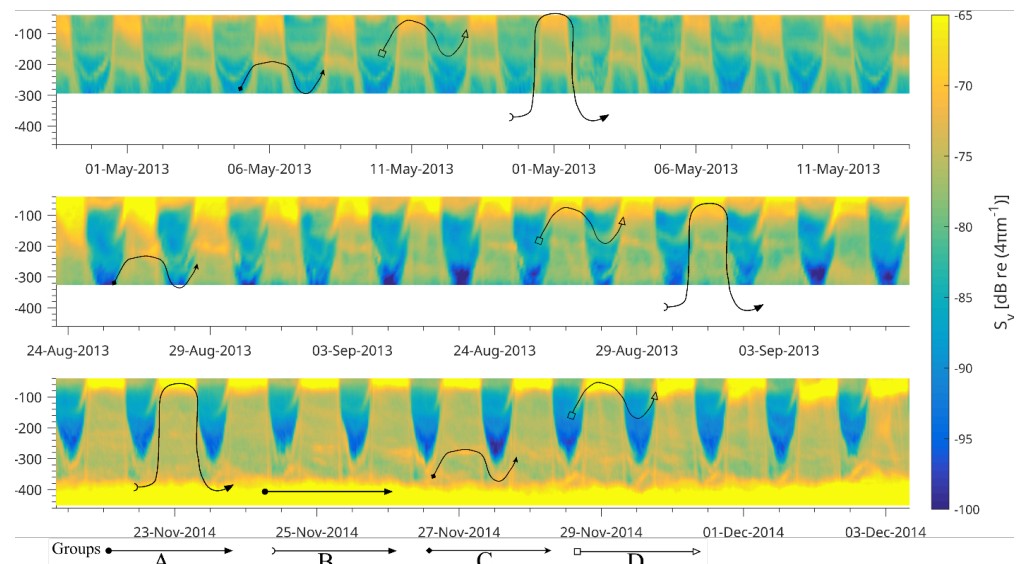

**Figure 6: Hand drawn trails of $S_v$ attributed to groups of planktonic and micro-nectonicorganisms.**



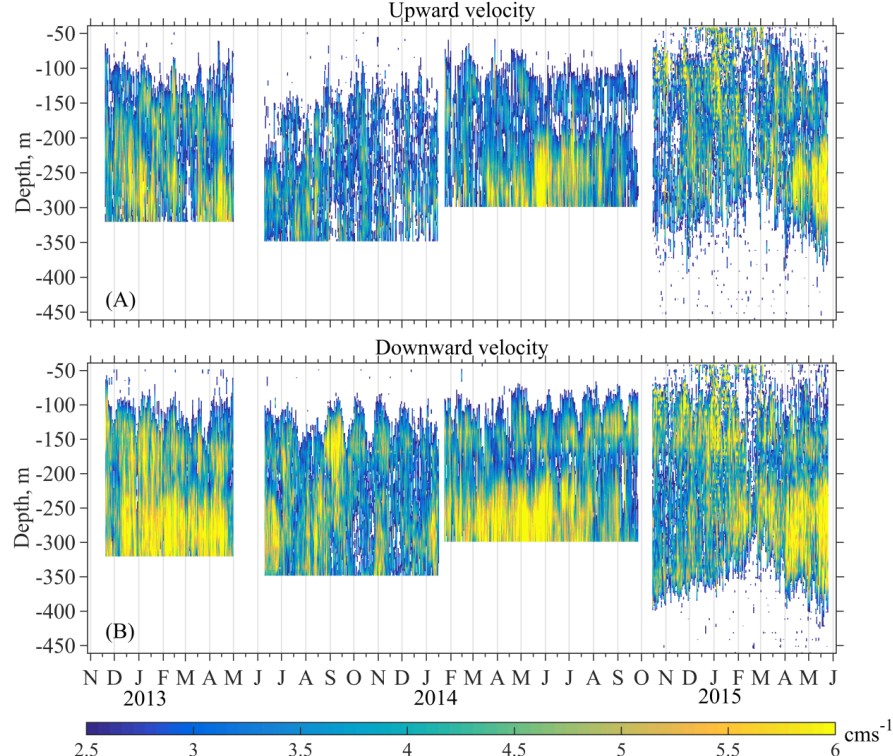

**Figure 7: Large upwards (A) and downwards (B) speed, attributed to the migration of zooplankton.**




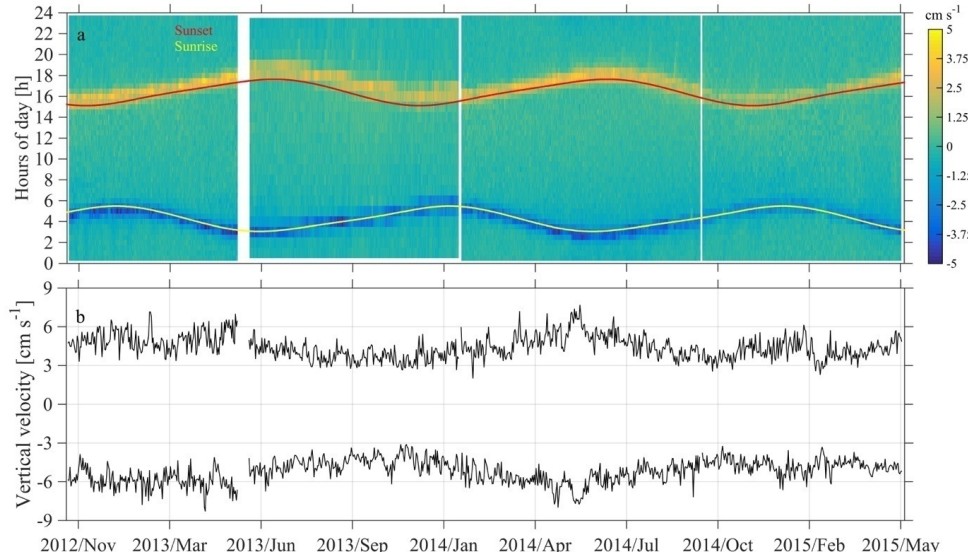

**Figure 8: Instantaneous depth averaged vertical velocities of daily segments of ADCP measurements (a) between 350 m and 50 m, following Jiang et al. (2007). Average of the three largest speed measurements per day (b). Sunrise and sunset times are superimposed.**





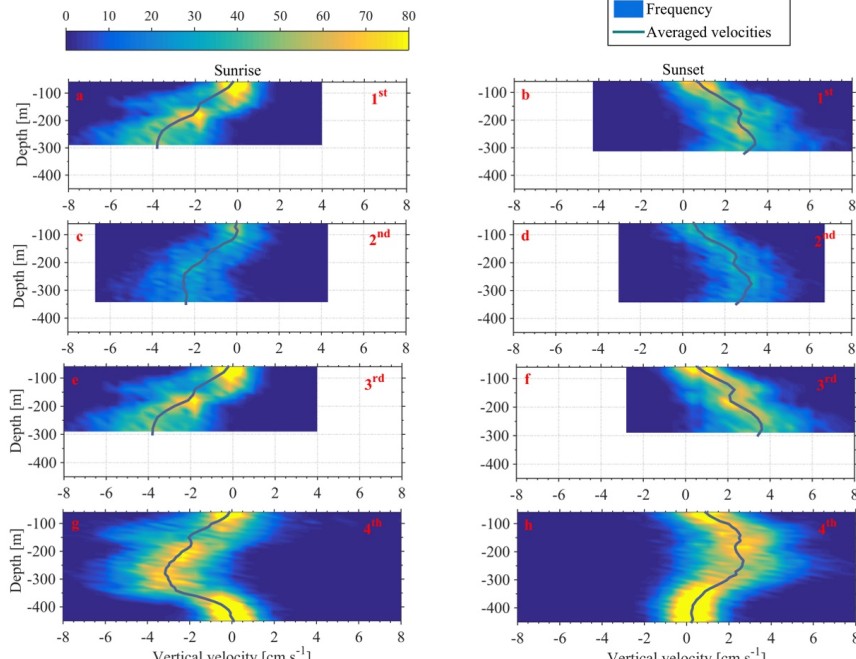

**Figure 9: Depth distributions of the vertical velocities, measured 1 h before and 1 h after sunset and sunrise, are shown. The time average velocity is superimposed. Each row of panels refers to one deployment (1st, 2nd, 3rd, 4th). The first column of panels corresponds to sunrise and the second column to sunset.**





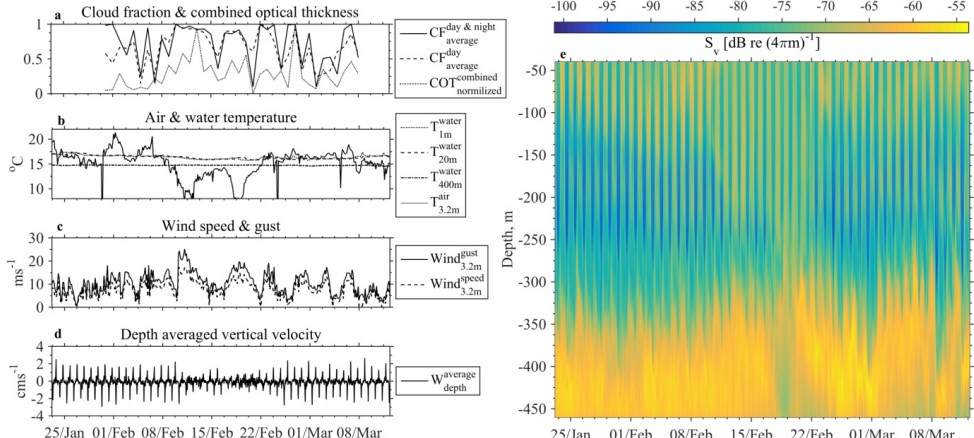

**Figure 10: Cloudiness (a), air and water temperature (b) and wind conditions (c) are examined in comparison to depth averaged vertical velocities (d) and backscatter coefficient (e) during February 2015.**



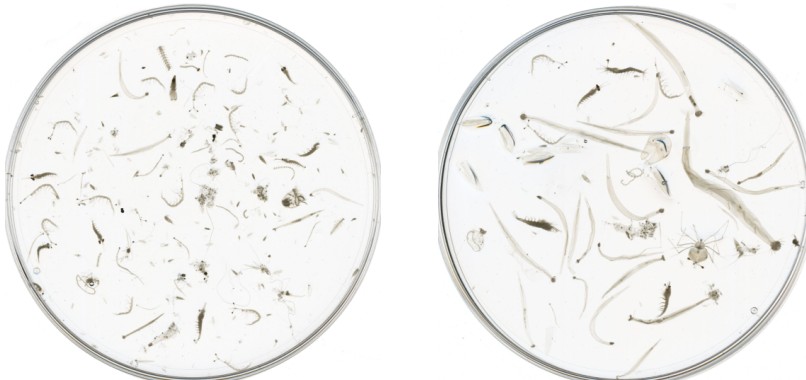

**Figure 11: Image of the larger organisms captured with a 500 μm mesh-size net in the 0-350 m layer at the E1-M3A location in December 2013 at mid-day (petri dish diameter is 8 cm).**