# Peer review of "ADCP observations of migration patterns of zooplankton in the Cretan Sea"

_Ocean Science, 2018_

## Referee Comment (RC1) · Anonymous Referee #1 · 17 Mar 2018

General comments

This manuscript presents the vertical distribution and migration of macro-zooplankton in the open Cretan Sea as derived from a 75 kHz ADCP dataset that covers a period of 30months with four deployments. The topic falls within the scope of Ocean Science and might constitute a valuable contribution to the Special Issue "Coastal marine infrastructure in support of monitoring, science, and policy strategies". The use of ADCP as a non-invasive method to infer the zooplankton distribution has been demonstrated by previous papers published on the subject in other oceanographic regions. The novelty of this study is that it applies the ADCP method for this purpose for the first time in the Eastern Mediterranean, in particular, in the Cretan Sea, a very dynamic and crucial area for the circulation of the EMed. Unfortunately, the first three ADCP deployments

were shallower (down to 300 m depth) than the fourth one (450 m) and missed some interesting zooplankton features. Despite the above mentioned positive aspects, the manuscript is not clear and focused enough to meet the Ocean Science standards. The manuscript should be more neat and concise to gain in clarity and needs a profound revision to solve the numerous problems I see in the various sections and have detailed in my specific comments. The main limit of this work is the lack of macro-zooplankton data collected with vertical tows that could have helped substantially to interpret the ADCP profiles. The experimental set up was clearly designed on the current measurements and not on zooplankton analysis, which was decided successively (at least, this appears from the content of this manuscript). Other papers have analysed ADCP data for inferring on zooplankton vertical migration without parallel zooplankton sampling, but this does not justify the critical limit determined by this choice.

Specific comments

Introduction needs to be carefully revised because the issues presented are not well linked and not clear in some parts. Topics like: the biological pump, the role of zooplankton vertical migration, the Eastern Mediterranean and the Cretan Sea are not present adequately. The biological pump should be introduced in the first sentence and expanded in the second one. The primary production is not linked to the following paragraph and can be removed because not developed further in this section. To be precise, the biological data from midwater are not lacking but they are few in comparison with epipelagic layers (L22-27, pg.1). The vertical migration of zooplankton comes out of the blue (L30); this topic relevant for this study should be properly introduced and the related papers in the Med should be cited. After that, it should be provided the info on zoo vertical migration available for the EMed and explained the "different migrating strategies" emerged from previous papers. The last part of pg. 1 is quite confused and should be rewritten. The Cretan Sea should be presented in a clear way; it is necessary to provide a brief description of the Sea with basic info on characteristics of hydrology, biochemistry and zooplankton in the area before claiming that it is representative of a wider EMed area. The second part of Introducion looks better and requires minor changes, apart the two following ones. 1-The study by Cardini et al. (2003) is misplaced here, in my opinion. If the present study has been stimulated by the hypothesis by Cardini et al. (same place, same ADCP) on zooplankton vertical migration, the rationale and the aim of the present study should be presented first and Cardini after (e.g., this study aimed at…..testing the hypothesis by ….). If the two studies were not related in any case, the citation should be removed from Introduction and used in Discussion. 2- The last sentence on carbon cycle should be removed: considerations on this topic must be treated with caution and not here, because the present study does not provide results on carbon data.

Methods The experimental set up was not designed to properly interpreting the ADCP data in relation to macro zooplankton distribution because of the lack of parallel zooplankton sampling. The zooplankton sampling done with a 200 $\mu$m net at monthly frequency in the upper 100 m and the single oblique tow performed with a Bongo net (330 $\mu$m and 500 $\mu$m) in Dec 2013 in the 0-500 m layer were useless for interpreting the ADCP data of this work and, in fact, those zooplankton data are not presented. Therefore, this part (pg5, L7-14) should be removed. Similarly, the downward looking 400kHz ADCP data were not used for the present work, so this part should be removed too (pg4, L34-36). The sampling and analytical methods for chlorophyll concentrations in Figure 4 are not presented. I do not understand how the burst velocity may help identifying the optimal sampling strategy for zooplankton; probably you mean "the optimal cell extension for the most appropriate recording of zooplankton signals"? Or something else? This part must be clarified (pg.3, L16). The affirmation on pg. 4, L3 is based on statistical test? Expand the explanation.

The Results section needs a substantial revision and a more synthetic rewriting. The results are interesting, but their too long, detailed and sometimes repetitive description leads to lose the focus and weaken the main messages. This section is definitely too long (9 pages) and repetitive and I found it heavy to read. It is presented in subsections, but it is not clear enough because the topics cross different sections, the figures are continuously cited back and forth (not in the numerical order for figs 7,9,10,8) and this creates confusion and tiredness in the reader. Links among figures are commented and interpretations are mixed with mere objective results. Links, interpretation, and comments should go all in Discussion to provide the reader with a neat and clear overview of the study. I suggest to follow the typical simple way of result presentation, i.e. describing objectively the data showed by the figures, in the right succession, without anticipating interpretation. This is not a "Results and Discussion" section. Part 3.1 on Environmental conditions is confusing; it is not clear what are the general typical conditions of the area as emerged from previous works and what are the results from the present work. There should be a separated section, placed in Methods, reporting the general characteristics of the area. The use of the past tense would help clarifying what are the results of the present study. The "constant presence of a deep layer of scatterers" (pg8, L11) was actually recorded only in the fourth deployment because deeper down to 450 m, so it cannot be defined "constant". The affirmation that zooplankton feed at certain depths and hours (e.g. pg.9, L8 and somewhere else) should be changes in likely supposition; the present study does not demonstrate any feeding activity, which might only be hypothesized as (one of the) possible explanation to the vertical zooplankton displacement inferred by ADCP backscatter. There are other strict affirmations that should be changed in interpretative suppositions/hypotheses (e.g., pg.10, L7-12). Many of the migrant macro-zooplankton animals are carnivorous. This kind of interpretation should be presented with more caution.

Discussion contains some repetitions of Results; the two sections should be rewritten in parallel to separate objective results from interpretation and discussion. This would make the manuscript much clearer and more interesting and pleasant to read. Discussion should be also organized in sub-sections to address the different aspects of this study. Light is interpreted as "the main factor affecting zooplankton migration" (pg.14, L28); it can be a triggering signal, a factor acting directly or indirectly on individual animals, or on swarms. This interpretation should be more exhaustively expanded by

using information from the literature on light perception by zooplankton groups. The last part of Discussion (pg.17, from L17) is dedicated to comments on the limits of this work, which are quite heavy indeed and should have been addressed at the beginning of the study, planning a more appropriate in-field-experimental design. It is right and honest to discuss on the limits, but this section should be closed with positive conclusions highlighting how this work contributes to increase the knowledge on zooplankton in the Cretan Sea. I am positive that this work might be useful to Mediterranean zooplanktonlogists, but the authors should convince better the readers.

Technical corrections -The past tense must be used for presenting the results of this study, not the present tense. -Zooplankton, as collective name, need plural verb. -Population must be removed or replaced by "assemblages" when associated to zooplankton because "population" has to be used only referring to species. -"Sampling" should be replaced by "recording" in relation to ADCP data; sampling is properly used for collection of samples (e.g. pg.3, L19-24 and somewhere else) -Some units are often written incorrectly with missing space throughout the manuscript (e.g., ms-1, cms-1 instead of the correct m s-1, cm s-1). -Velocity and speed are used indifferently (e.g. in the caption of Fig. 8); better using one of the two throughout the whole manuscript. Pg.1, L28: Basin instead of "Part" Pg2, L4: "several meters" length pertains to long chains of gelatinous zooplankton like salps, for example; it is written wrong here because it seems that some medusae (jellyfish) are several meters large, that's not true in the Med. Pg2, L10: delete "However" L10-13: the two sentences repeat the same concept (ADCP detects zooplankton) and should be merged. L35: "south Aegean Sea" must be indicated the first time the Cretan Sea is mentioned in Introduction, not here. L37: "Water-plankton sampling" is an awkward expression; it should be Niskin sampling for chlorophyll measurements (or phytoplankton, microzooplankton) or net tows for zooplankton sampling. L39: 6 months not 7, from 15/11/2012 till 20/5/2015 Pg3, L16: "behaviour" can be many things, sospecify "vertical movement" or migration Pg4, L18, L20:"pieces" should be replaced by "datasets" or "sections" or a more appropriate term. L19: it's not really a long-term, better specify seasonal and interannual variability L33, 34: add "depth" after 100 m and 250 m (check throughout the manuscript) Pg5,L1,2: "M" must be replaced by "months Pg6, L21: "typical values" are averages, medians? Clarify L25: zooplankton has not been introduced yet; this link between results should be moved to Discussion. Pg7, L5; "Figure" is repeated L7: as reported by Cardin et al. (2003). Pg8, L9: . . .four ADCP deployments. . . Pg9, L10: the daily data are embedded in the graphs of Fig.5 but the daily resolution is not visible. Pg10, L8: might be due, not "must" Pg14, L26: I do not understand how groups A and B "are coupled in a behavioural relation"; it should be rephrased and clarified.

Figures Fig.2- the research vessels should not be mentioned in the figure captions; the upper values of y-axis in panels a, d, e, should be indicated. Fig.3-the unit "m" is missing on the y-axis Fig.4- The "grey dotted lines" are barely visible in the upper panel. I see only a single value in blue, and not a range above each cast; is it the max value? Fig.5- The explanation of the black and yellow circles in the caption is wrong, it's the opposite. Repeat here the explanation of the triangles and bars. Fig.6- The day and night times should be indicated in this figure as referred to in the text. Fig.8- I do not see the 4 hours indicated in the text (pg.10, L26); the hours should be indicated in the graph. I suppose that "largest speed measurements" mean actually "highest speed values"(or velocity?). Fig.9- The velocity unit on the reference coloured bar is missing. For each deployment, the data displayed are time-averaged, I suppose. This has to be clearly indicated. Fig. 10- The events of harsh weather reported in section 3.4 should be indicated on panel e). Fig.11- It is nice but not necessary because it shows a qualitative snapshot (1 sample) of the zooplankton community captured with a 500 $\mu$m Bongo net, not useful enough to interpret the ADCP data.

References The Mann&Lazier book is reported as 2005 edition here and 1991 in the text (pg.2, L1).

---

## Referee Comment (RC2) · Anonymous Referee #2 · 19 Mar 2018

The paper "ADCP observations of migration patterns of zooplankton in the Cretan Sea" by E. Potiris et al. presents the analysis of about 2.5 years of acoustic data from several deployment of an ADCP (RDI, 75 kHz) on a sub surface mooring in the Cretan Sea in order to infer the migration patterns of the zooplankton population in that area. The analysis is based on raw backscatter acoustic data from ADCP and ancillary data collected by a fixed open ocean observatory (E1-M3A), during cruises or from external data centers. An interesting aspect of the paper is the demonstration that ADCP data collected for other "standard" purposes (namely investigation of currents in the water column) can be also used to gather biological information (e.g., vertical migration of zooplankton). This might give new life to already existing dataset, not completely exploited yet. The paper is very well written, clear, easy to read and to understand.

[Figure]

The paper starts describing the processing method and the data, the environmental conditions (both during the experiment period and with a climatic perspective). The discussion about the backscattering data and the reasoning about the different groups of organisms is really amazing and nicely justified. Although, unfortunately, not fully supported by in-situ catchments, the authors were able to properly correlate acoustic data with biological information as well as with meteorological conditions. Just the latter analysis is one of the novel finding of the paper that is not usually in other biological-inspired works. The only significant migratory pattern evidenced is the normal one, with zooplankton species going toward deeper layers during the day and going upward toward the surface to feed during the night. The contribution of moon phases, as already evidenced by other cited studies, was also taken into account and discussed. The distinction of the four groups of zooplankton (A,B,C,D) is based only on a visual analysis of Sv values; maybe it could be useful to analyze the vertical velocities at different depths to better identify the migratory pattern of each group. I recommend the publication of the paper after minor revisions.

Specific comments. Throughout the manuscript several symbols ">", "<", "~" are used. I would suggest to replace such symbols with words, i.e. greater, lower, about etc. Title of section 2 and 2.2 have some capitalized letter. Please make them uniform with the other titles. Page 5, line 10. The unit NM (nautical mile?) seems to refer to a speed. Likely it should be change to knot. Page 5, table 2. It includes the pH parameter which is not mentioned in the text nor used in the description of the ancillary data. I think it is worth discarding it. Page 7, line 5. Correct the reference to Figure 3: "Figure 3Figure b&c". Page 6, figure 2. Although the background color of the plots in the third row is grey, the used color scale doesn't allow to appreciate all lines. Especially those plotted in white are almost invisible. Please, consider to change the color scale. Page 8, figure 5. The labels above the plots and the caption seems not to agree with respect to the full moon and new moon. Page 12, figure 8. Caption is not clear and, for example, it might be re-arranged as follows: (a) Instantaneous depth averaged vertical velocities of daily segment of ADCP measurements between 350 m and 50 m, following Jiang

et al. (2007). Sunrise and sunset times are superimposed. (b) Average of the three largest speed measurements per day. Page 13, figure 9. The unit of the color bar is missing.
* * *

---

## Author Comment (AC1) · 30 May 2018

REPLIES TO REVIEWER 1

We thank the reviewer for carefully reading the manuscript and providing useful comments.

This manuscript presents the vertical distribution and migration of macro-zooplankton in the open Cretan Sea as derived from a 75 kHz ADCP dataset that covers a period of 30months with four deployments. The topic falls within the scope of Ocean Science and might constitute a valuable contribution to the Special Issue "Coastal marine infrastructure in support of monitoring, science, and policy strategies". The use of ADCP as a non-invasive method to infer the zooplankton distribution has been demonstrated

by previous papers published on the subject in other oceanographic regions. The novelty of this study is that it applies the ADCP method for this purpose for the first time in the Eastern Mediterranean, in particular, in the Cretan Sea, a very dynamic and crucial area for the circulation of the EMed. Unfortunately, the first three ADCP deployments were shallower (down to 300 m depth) than the fourth one (450 m) and missed some interesting zooplankton features. Despite the above mentioned positive aspects, the manuscript is not clear and focused enough to meet the Ocean Science standards. The manuscript should be more neat and concise to gain in clarity and needs a profound revision to solve the numerous problems I see in the various sections and have detailed in my specific comments. The main limit of this work is the lack of macro-zooplankton data collected with vertical tows that could have helped substantially to interpret the ADCP profiles. The experimental set up was clearly designed on the current measurements and not on zooplankton analysis, which was decided successively (at least, this appears from the content of this manuscript). Other papers have analysed ADCP data for inferring on zooplankton vertical migration without parallel zooplankton sampling, but this does not justify the critical limit determined by this choice.

SPECIFIC COMMENTS

Introduction needs to be carefully revised because the issues presented are not well linked and not clear in some parts. Topics like: the biological pump, the role of zooplankton vertical migration, the Eastern Mediterranean and the Cretan Sea are not present adequately. The biological pump should be introduced in the first sentence and expanded in the second one. The primary production is not linked to the following paragraph and can be removed because not developed further in this section. To be precise, the biological data from midwater are not lacking but they are few in comparison with epipelagic layers (L22-27, pg.1). The vertical migration of zooplankton comes out of the blue (L30); this topic relevant for this study should be properly introduced and the related papers in the Med should be cited. After that, it should be provided the info on zoo vertical migration available for the EMed and explained the "different

migrating strategies" emerged from previous papers. The last part of pg. 1 is quite confused and should be rewritten. The Cretan Sea should be presented in a clear way; it is necessary to provide a brief description of the Sea with basic info on characteristics of hydrology, biochemistry and zooplankton in the area before claiming that it is representative of a wider EMed area.

Reply: The first part of the introduction was restructured in the following order: brief description of biological pump (without mention of primary production), diel vertical migration (DVM) description (including patterns, related factors), DVM studies by ADCP, DVM studies in the Western and Eastern Mediterranean Sea (including migrating strategies), description of Cretan Sea (hydrology, biochemistry and zooplankton), Mediterranean DVM studies by ADCP. We specified the gaps of knowledge (instead of gap of data) regarding midwater depths. New references were added to support the above description. The reasons why the Cretan Sea is representative of a wider area was more developed.

The second part of Introduction looks better and requires minor changes, apart the two following ones. 1-The study by Cardini et al. (2003) is misplaced here, in my opinion. If the present study has been stimulated by the hypothesis by Cardini et al. (same place, same ADCP) on zooplankton vertical migration, the rationale and the aim of the present study should be presented first and Cardini after (e.g., this study aimed at. . ...testing the hypothesis by . . ..). If the two studies were not related in any case, the citation should be removed from Introduction and used in Discussion.

Reply: The two studies were related. The last paragraph of the Introduction section was rewritten to reflect this fact as suggested.

The last sentence on carbon cycle should be removed: considerations on this topic must be treated with caution and not here, because the present study does not provide results on carbon data.

Reply : The sentence was removed from the text.

Methods The experimental set up was not designed to properly interpreting the ADCP data in relation to macro zooplankton distribution because of the lack of parallel zoo plankton sampling. The zooplankton sampling done with a 200 $\mu$m net at monthly frequency in the upper 100 m and the single oblique tow performed with a Bongo net (330 $\mu$m and 500 $\mu$m) in Dec 2013 in the 0-500 m layer were useless for interpreting the ADCP data of this work and, in fact, those zooplankton data are not presented. Therefore, this part (pg5, L7-14) should be removed.

Reply: The paragraph "At the E1-M3A zooplankton. . . . . . mentioned briefly in the discussion." was removed.

Similarly, the downward looking 400kHz ADCP data were not used for the present work, so this part should be removed too (pg4, L34-36).

Reply: The 400kHz ADCP data were used in Figure 3, b. The data are presented at pg. 6 L26 and pg.7 L2-4.

The sampling and analytical methods for chlorophyll concentrations in Figure 4 are not presented.

Reply: The chlorophyll concentrations were measured with the WETLABS ECO FLNTU sensors which were mounted on the profiling 19plus and moored 16plus CTDs. The above information was added to the text.

I do not understand how the burst velocity may help identifying the optimal sampling strategy for zooplankton; probably you mean "the optimal cell extension for the most appropriate recording of zooplankton signals"? Or something else? This part must be clarified (pg.3, L16).

Reply: The burst velocity was not properly defined in the text. The paragraph containing the burst velocity definition was rephrased as follows: "One parameter used to potentially identify the optimal cell extension and sampling interval for the most appropriate recording of zooplankton signals was the hereafter defined burst speed. The

burst speeds of each cell are defined as the highest and lowest vertical velocity measurements respectively during a time period of one day. The velocity measurement inside a cell over the sampling interval is the result of the averaging of several pings. As the sampling interval increases and/or the cell extension decreases and/or the actual zooplankton speed increases, we expect the actual zooplankton speed to be underestimated because zooplankton will not be inside the cell throughout the duration of the measurement but only during a fraction of it. The largest underestimation is expected when the actual zooplankton migrating speed is maximum. Thus, comparison of upward and downward burst velocities between deployments at depths around 250 m were used to identify the most appropriate sampling scheme."

The affirmation on pg. 4, L3 is based on statistical test? Expand the explanation.

Reply: The comment refers to the phrase "The range of the cells used (10 - 20 m) did not affect the burst velocity and the average velocity measurements." The sampling rate of deployments No 1, 3 and 4 was the same (30 min). If a bias was caused to the burst and average velocities due to the cell extension (as explained in reply to the previous comment) then choosing a smaller cell extension should result in smaller burst and average velocities. Smaller cell extensions of deployments with the same sampling interval did not result in smaller velocities, although the conclusion was based on visual inspection of burst and average velocities plots and not on a statistical test. The following phrase was added to the text: "The range of the cells used (10 - 20 m), on the other hand, did not affect the burst speed and the average velocity measurements. Based on visual inspection, smaller cell extension during the 1st, 3rd and 4th deployments (30 min sampling interval) did not result in smaller burst and average velocities."

The Results section needs a substantial revision and a more synthetic rewriting. The results are interesting, but their too long, detailed and sometimes repetitive description leads to lose the focus and weaken the main messages. This section is definitely too long (9 pages) and repetitive and I found it heavy to read.

Reply: Results section was reduced by about one page, some parts were rewritten, and repetitions were removed.

It is presented in subsections, but it is not clear enough because the topics cross different sections, the figures are continuously cited back and forth (not in the numerical order for figs 7,9,10,8) and this creates confusion and tiredness in the reader.

Reply: The back and forth citation of figures was reduced to a minimum. Figures' citations are now in numerical order.

Links among figures are commented and interpretations are mixed with mere objective results. Links, interpretation, and comments should go all in Discussion to provide the reader with a neat and clear overview of the study. I suggest to follow the typical simple way of result presentation, i.e. describing objectively the data showed by the figures, in the right succession, without anticipating interpretation. This is not a "Results and Discussion" section.

Reply: Interpretations, links among figures and their comments were removed from the Results section.

Part 3.1 on Environmental conditions is confusing; it is not clear what are the general typical conditions of the area as emerged from previous works and what are the results from the present work. There should be a separated section, placed in Methods, reporting the general characteristics of the area.

Reply: A "Hydrology of the Cretan sea" subsection was added under the "Methodology" section. No references are now present in Part 3.1 of the manuscript.

The use of the past tense would help clarifying what are the results of the present study.

Reply: The results were written in past tense to clarify the results.

The "constant presence of a deep layer of scatterers" (pg8, L11) was actually recorded

only in the fourth deployment because deeper down to 450 m, so it cannot be defined "constant".

Reply: The term "constant" was removed from the text.

The affirmation that zooplankton feed at certain depths and hours (e.g. pg.9, L8 and somewhere else) should be changes in likely supposition; the present study does not demonstrate any feeding activity, which might only be hypothesized as (one of the) possible explanation to the vertical zooplankton displacement inferred by ADCP backscatter.

Reply: Changed in the text as suggested.

There are other strict affirmations that should be changed in interpretative suppositions/hypotheses (e.g.,pg.10, L7-12). Many of the migrant macro-zooplankton animals are carnivorous. This kind of interpretation should be presented with more caution.

Reply: The last paragraph of section 3.2 was removed from the manuscript and rewritten in a less affirmative way as suggested by the reviewer

Discussion contains some repetitions of Results; the two sections should be rewritten in parallel to separate objective results from interpretation and discussion. This would make the manuscript much clearer and more interesting and pleasant to read.

Reply: Repetition of results were removed from Discussion section and parts of it were rewritten.

Discussion should be also organized in sub-sections to address the different aspects of this study.

Reply: The Discussion section is now organised in sub-sections.

Light is interpreted as "the main factor affecting zooplankton migration" (pg.14, L28); it can be a triggering signal, a factor acting directly or indirectly on individual animals, or on swarms. This interpretation should be more exhaustively expanded by using

information from the literature on light perception by zooplankton groups.

Reply: The interpretation was expanded using information from the literature as suggested.

The last part of Discussion (pg.17, from L17) is dedicated to comments on the limits of this work, which are quite heavy indeed and should have been addressed at the beginning of the study, planning a more appropriate in-field-experimental design. It is right and honest to discuss on the limits, but this section should be closed with positive conclusions highlighting how this work contributes to increase the knowledge on zooplankton in the Cretan Sea. I am positive that this work might be useful to Mediterranean zoo planktonlogists, but the authors should convince better the readers.

Reply: The paragraph explaining the limitations of the methodology was moved to the last part of the Methods Section under the Limitations subsection.

Technical corrections The past tense must be used for presenting the results of this study, not the present tense.

Reply: The results were written in past tense to clarify the results.

Zooplankton, as collective name, need plural verb.

Reply: Corrected to plural.

Population must be removed or replaced by "assemblages" when associated to zoo plankton because "population" has to be used only referring to species.

Reply: Population was replaced by assemblages.

"Sampling" should be replaced by "recording" in relation to ADCP data; sampling is properly used for collection of samples (e.g. pg.3, L19-24 and somewhere else)

Reply: "Sampling" was replaced "recording" as suggested.

Some units are often written incorrectly with missing space throughout the manuscript

(e.g., ms-1, cms-1 instead of the correct m s-1, cm s-1).

Reply: Units were corrected.

Velocity and speed are used indifferently (e.g. in the caption of Fig. 8); better using one of the two throughout the whole manuscript.

Reply: Speed was changed to velocity in the caption if Fig. 8 as it was used incorrectly. For the rest of the manuscript, speed was used to indicate the absolute vertical velocity or the magnitude of horizontal velocity (currents and wind).

Pg.1, L28: Basin instead of "Part" Pg2, L4: "several meters" length pertains to long chains of gelatinous zooplankton like salps, for example; it is written wrong here because it seems that some medusae (jellyfish) are several meters large, that's not true in the Med.

Reply: "Part" was replaced by "basin". "several meters" was removed as the whole introduction was rewritten.

Pg2, L10: delete "However" L10-13: the two sentences repeat the same concept (ADCP detects zooplankton) and should be merged.

Reply: "However" was deleted.

L35: "south Aegean Sea" must be indicated the first time the Cretan Sea is mentioned in Introduction, not here.

Reply: "south Aegean Sea" was moved to the abstract.

L37: "Water-plankton sampling" is an awkward expression; it should be Niskin sampling for chlorophyll measurements (or phytoplankton, microzooplankton) or net tows for zooplankton sampling.

Reply: The sentence was replaced by "The observing effort is complemented by monthly R/V cruises to perform CTD casts (temperature, salinity, fluorescence) and

net tows (zooplankton)."

L39: 6 months not 7, from 15/11/2012 till 20/5/2015

Reply: Corrected to "six months".

Pg3, L16: "behaviour" can be many things, so specify "vertical movement" or migration

Reply: The whole paragraph was rephrased.

Pg4, L18, L20:"pieces" should be replaced by "datasets" or "sections" or a more appropriate term.

Reply: "Pieces" was replaced by "datasets".

L19: it's not really a long-term, better specify seasonal and interannual variability

Reply: "long-term" replaced by "seasonal and interannual".

L33, 34: add "depth" after 100 m and 250 m (check throughout the manuscript)

Reply: "depth" added throughout the manuscript.

Pg5,L1,2: "M" must be replaced by "months "

Reply: "M" replaced by months.

Pg6, L21: "typical values" are averages, medians? Clarify

Reply: "typical values. . ." replaced by "average wind stress is 0.82 Pa"

L25: zooplankton has not been introduced yet; this link between results should be moved to Discussion.

Reply: Removed as suggested.

Pg7, L5; "Figure" is repeated

Reply: "Figure" was removed.

L7: as reported by Cardin et al. (2003).

Reply: Rephrased as suggested.

Pg8, L9: . . .four ADCP deployments. . .

Reply: Rephrased as suggested.

Pg9, L10: the daily data are embedded in the graphs of Fig.5 but the daily resolution is not visible.

Reply: The "... a diel. . ." was removed.

Pg10, L8: might be due, not "must"

Reply: Rephrased as suggested.

Pg14, L26: I do not understand how groups A and B "are coupled in a behavioural relation"; it should be rephrased and clarified.

Reply: The paragraph containing the phrase "are coupled in a behavioural relation" was removed from the manuscript.

Figures Fig.2- the research vessels should not be mentioned in the figure captions; the upper values of y-axis in panels a, d, e, should be indicated.

Reply: The research vessels were removed from the caption. The upper and lower values of the y-axis in panels a, d, e are now indicated.

Fig.3-the unit "m" is missing on the y-axis

Reply: Depth [m] was added to the y-axis.

Fig.4- The "grey dotted lines" are barely visible in the upper panel. I see only a single value in blue, and not a range above each cast; is it the max value?

Reply: The "grey dotted lines" was changed to "grey lines". The light grey color for the minimum cast value was chosen because otherwise the figure would be cluttered,

and the normalized chlorophyll data would not be easily visible. Above each cast there are two numbers: the original chlorophyll range of the cast (in blue) and the original minimum chlorophyll value (in red). The words "Min" and "Range", colored red and blue respectively, are shown to the left of the colored numbers.

Fig.5- The explanation of the black and yellow circles in the caption is wrong, it's the opposite. Repeat here the explanation of the triangles and bars.

Reply: The explanation of the black and yellow circles was corrected, and the explanation of the triangles and bars was added to the caption.

Fig.6- The day and night times should be indicated in this figure as referred to in the text.

Reply: The day and night times were added to the figure.

Fig.8- I do not see the 4 hours indicated in the text (pg.10, L26); the hours should be indicated in the graph. I suppose that "largest speed measurements" mean actually "highest speed values"(or velocity?).

Reply: The hours indicated in the text were added to the figure. The caption was rephrased to "(a) Instantaneous depth averaged vertical velocities of daily segments of ADCP measurements between 350 m and 50 m, following Jiang et al. (2007). Sunrise and sunset times are superimposed. (b) Average of the three highest upwards and downwards speed values per day. The hours of fast zooplankton motion are also shown.".

Fig.9- The velocity unit on the reference coloured bar is missing. For each deployment, the data displayed are time-averaged, I suppose. This has to be clearly indicated.

Reply: The plots are histograms to visualise data density. The color bar show number of measurements inside each bin (the units were added next to the bar). The averaged velocities are also plotted with the thick lines. The legend was changed to clarify the figure. "The time average velocity at each depth is superimposed" was added to the

caption.

Fig. 10- The events of harsh weather reported in section 3.4 should be indicated on panel e).

Reply: The events of harsh weather were shaded on all panels. "Grey shaded areas denote the three harsh weather events referred in the text." was added to the caption.

Fig.11- It is nice but not necessary because it shows a qualitative snapshot (1 sample) of the zooplankton community captured with a 500 $\mu$m Bongo net, not useful enough to interpret the ADCP data.

Reply: The figure was removed from the manuscript. The corresponding text was modified in the discussion.

References The Mann&Lazier book is reported as 2005 edition here and 1991 in the text (pg.2, L1).

Reply: The text reference was changed to 2005.

REPLIES TO REVIEWER 2

We thank the reviewer for carefully reading the manuscript and providing useful comments.

The paper "ADCP observations of migration patterns of zooplankton in the Cretan Sea" by E. Potiris et al. presents the analysis of about 2.5 years of acoustic data from several deployment of an ADCP (RDI, 75 kHz) on a sub surface mooring in the Cretan Sea in order to infer the migration patterns of the zooplankton population in that area. The analysis is based on raw backscatter acoustic data from ADCP and ancillary data collected by a fixed open ocean observatory (E1-M3A), during cruises or from external data centers. An interesting aspect of the paper is the demonstration that ADCP data collected for other "standard" purposes (namely investigation of currents in the water column) can be also used to gather biological information (e.g., vertical migration

of zooplankton). This might give new life to already existing dataset, not completely exploited yet. The paper is very well written, clear, easy to read and to understand. The paper starts describing the processing method and the data, the environmental conditions (both during the experiment period and with a climatic perspective). The discussion about the backscattering data and the reasoning about the different groups of organisms is really amazing and nicely justified. Although, unfortunately, not fully supported by in-situ catchments, the authors were able to properly correlate acoustic data with biological information as well as with meteorological conditions. Just the latter analysis is one of the novel finding of the paper that is not usually in other biological- inspired works. The only significant migratory pattern evidenced is the normal one, with zooplankton species going toward deeper layers during the day and going upward toward the surface to feed during the night. The contribution of moon phases, as already evidenced by other cited studies, was also taken into account and discussed.

The distinction of the four groups of zooplankton (A,B,C,D) is based only on a visual analysis of Sv values; maybe it could be useful to analyze the vertical velocities at different depths to better identify the migratory pattern of each group.

Reply: Vertical velocities were analysed along with the backscatter coefficient at different depths. The analysis resulted in the update of some of our previous results. The updates were incorporated in the manuscript in the Results section.

SPECIFIC COMMENTS Throughout the manuscript several symbols ">", "<", "âĹij" are used. I would suggest to replace such symbols with words, i.e. greater, lower, about etc.

Reply: The mentioned symbols were replaced with words as suggested.

Title of section 2 and 2.2 have some capitalized letter. Please make them uniform with the other titles.

Reply: The titles were made uniform.

Page 5, line 10. The unit NM (nautical mile?) seems to refer to a speed. Likely it should be changed to knot.

Reply: The whole paragraph was removed as suggested by reviewer 1.

Page 5, table 2. It includes the pH parameter which is not mentioned in the text nor used in the description of the ancillary data. I think it is worth discarding it.

Reply: The pH parameter in the table was discarded as suggested.

Page 7, line 5. Correct the reference to Figure 3: "Figure 3Figure b&c".

Reply: The reference to Figure 3 is now "Figure 3, b & c".

Page 6, figure 2. Although the background color of the plots in the third row is grey, the used color scale doesn't allow to appreciate all lines. Especially those plotted in white are almost invisible. Please, consider to change the color scale.

Reply: The color scale was changed as suggested.

Page 8, figure 5. The labels above the plots and the caption seems not to agree with respect to the full moon and new moon.

Reply: The caption was corrected: "The yellow and black circles...".

Page 12, figure 8. Caption is not clear and, for example, it might be re-arranged as follows: (a) Instantaneous depth averaged vertical velocities of daily segment of ADCP measurements between 350 m and 50 m, following Jiang C2 et al. (2007). Sunrise and sunset times are superimposed. (b) Average of the three largest speed measurements per day.

Reply: Caption was re-arranged as "(a) Instantaneous depth averaged vertical velocities of daily segments of ADCP measurements between 350 m and 50 m, following Jiang et al. (2007). Sunrise and sunset times are superimposed. (b) Average of the three highest upwards and downwards speed values per day. The hours of faster zooplankton motion are also shown".

Page 13, figure 9. The unit of the color bar is missing.

Reply: Color bar units were added to the figure.